# SpikeGS: 3D Gaussian Splatting from Spike Streams with High-Speed Camera Motion

## ABSTRACT

Novel View Synthesis plays a crucial role by generating new 2D renderings from multi-view images of 3D scenes. However, capturing high-speed scenes with conventional cameras often leads to motion blur, hindering the effectiveness of 3D reconstruction. To address this challenge, high-frame-rate dense 3D reconstruction emerges as a vital technique, enabling detailed and accurate modeling of real-world objects or scenes in various fields, including Virtual Reality or embodied AI. Spike cameras, a novel type of neuromorphic sensor, continuously record scenes with an ultra-high temporal resolution, showing potential for accurate 3D reconstruction. Despite their promise, existing approaches, such as applying Neural Radiance Fields (NeRF) to spike cameras, encounter challenges due to the time-consuming rendering process. To address this issue, we make the first attempt to introduce the 3D Gaussian Splatting (3DGS) into spike cameras in high-speed capture, providing 3DGS as dense and continuous clues of views, then constructing **SpikeGS**. Specifically, to train SpikeGS, we establish computational equations between the rendering process of 3DGS and the processes of instantaneous imaging and exposing-like imaging of the continuous spike stream. Besides, we build a very lightweight but effective mapping process from spikes to instant images to support training. Furthermore, we introduced a new spike-based 3D rendering dataset for validation. Extensive experiments have demonstrated our method possesses the high quality of novel view rendering, proving the tremendous potential of spike cameras in modeling 3D scenes.

## CCS CONCEPTS

• **Computing methodologies** → **3D imaging**; **Computational photography**; *Motion capture*; *Reconstruction*; *Scene understanding*.

## KEYWORDS

View Synthesis, Dense 3D reconstruction, Spike Camera, Gaussian Splatting

## 1 INTRODUCTION

Novel View Synthesis (NVS) involves the generation of new, unseen 2D renderings of a viewpoint from a sequence of multi-view images of a given 3D scene. This task holds significant importance in the realm of 3D scene reconstruction topic, playing a crucial role in computer vision and imaging research. The introduction of Neural

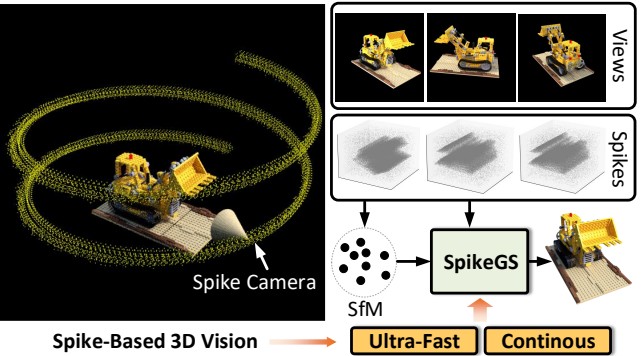

**Figure 1: Illustration of spike-based Novel View Synthesis (NVS). Spike cameras, with ultra-high-speed continuous imaging capability, capture dense views and overcome the blurring effects of exposure imaging. We make the first step to present Spike-based Gaussian Splatting (*SpikeGS*), proving the potential of spike cameras in real-time 3D Imaging.**

Radiance Fields (NeRF) [24] has particularly drawn attention to this field. NeRF combines implicit neural representations with volume rendering techniques, paving the way for innovative approaches in NVS. In recent years, there have been remarkable developments in NeRF-related technologies, including enhanced rendering methods for higher scene quality [33, 37], strategies tailored for handling more complex [22] and dynamic scenes [7, 27], and techniques for image deblurring [20]. NeRF learns the continuous volumetric density and color that implicitly represents scenes by training a Multi-Layer Perception (MLP) network. However, rendering a new viewpoint still requires a great amount of sampling and integration through MLP, imposing limitations on rendering speed.

**Why 3DGS standing out?** Recently, 3D Gaussian Splatting (3DGS) [10] has been proposed to achieve real-time render speed and more reliable performance. Different from NeRF which models scenes implicitly, 3DGS represents scenes explicitly with a series of 3D Gaussians, which is initialized by Structure-from-Motion (SfM) [32]. Each Gaussian is parameterized by the mean position, the full 3D covariance matrix, the opacity, and its color. 3DGS projects 3D Gaussians to the 2D image plane with the differentiable Gaussian rasterization, which makes it able to be optimized by gradients of 3D Gaussians. It achieves very short training time and rendering speed and possesses great potential on NVS.

**Degradation of 3DGS with camera motion.** Despite the remarkable efficacy demonstrated by 3DGS methodologies, their performance encounters inherent limitations imposed by the procedural characteristics of traditional exposure-based photo capturing. The conventional cameras, predicated by discrete exposure mechanisms, capture each frame within a predetermined temporal exposure window. This paradigm introduces a significant constraint

when the image sequence intended for training 3DGS is subjected to blurring attributable to the high-velocity capture process. Such conditions lead to two profound detriments of the 3DGS framework. Firstly, the prerequisite quality of the initial point cloud essential for 3DGS is severely compromised. Training high-quality 3D Gaussians requires accurate assumptions about camera poses, which is difficult to achieve in some real-world scenarios. Secondly, the blurry images would affect optimizing the covariance matrix of the 3D Gaussians [14]. Moreover, the intrinsic interval between successive frames in traditional cameras entails a temporal void during which no visual information is captured. This hiatus in data acquisition may result in the omission of pivotal viewpoint information for scenes demanding dense perspective sampling for high-grade rendering, thereby adversely affecting the integrity of novel view synthesis. If we can accurately capture dense and continuous views, the performance of 3D reconstruction may make progress.

**Introducing spike cameras for 3D reconstruction.** The spike camera represents a novel class of neuromorphic visual sensors, boasting advantages such as ultra-high temporal resolution and a higher dynamic range. Inspired by the mechanism of the fovea in the retinas of primates, each unit on a spike camera asynchronously and continuously receives photons and accumulates photoelectric current, immediately emitting a spike when the voltage reaches a preset threshold. Event cameras [2, 18, 25] are also kind of neuromorphic cameras which also possess high temporal resolution. Several studies integrate event cameras with NeRF for NVS. However, events encode the change of light and do not have absolute intensity information. **spike cameras** encode the absolute light intensity of a scene at extremely high speeds, which reduces the significance of exposure time. This characteristic naturally mitigates the presence of blur and alleviates the speed requirements for the camera during the shooting process. Existing studies [43] have proved the temporal and spatial completeness of spikes in 2D reconstruction. In 3D scenes, spikes provide a denser and more continuous set of viewpoints. We believe that spike cameras hold tremendous potential for 3D scene reconstruction. Recently, pioneering work has been carried out with SpikeNeRF [46], demonstrating the feasibility of using spikes in modeling 3D scenes. However, SpikeNeRF faces several challenges: first, due to its complex spike simulation process, both training and rendering speeds are suboptimal; second, its training requires noise estimation to be recalibrated for different scenes, indicating a lack of adaptability; third, it fails to leverage the high temporal resolution advantage of spike cameras fully. This paper aims to fully exploit the high-speed and continuous imaging advantages of spiking cameras, constructing a spike-based 3D Gaussian Splatting model for the first time, and overcoming the limitations of training 3DGS on traditional RGB sequences.

**What attempts we have made for spike-based GS?** In this work, we make the first attempt to introduce the 3D Gaussian Splatting (3DGS) into spike cameras in high-speed capture, providing 3DGS as great supervision signals and constructing **SpikeGS**.

To be specific, we first build the framework of **SpikeGS** based on continuous spikes. we focus on two characteristics to assist the training of high-quality 3D scenes from the fast-moving spike camera: *Instantaneous Imaging from spikes*, and *Exposing-like Imaging from spikes*. On the one hand, to meet the instantaneous imaging assumption in 3DGS rendering, we aim to establish a 'simple but

effective' mapping from continuous spikes to instant images, which can offer good signals for supervising the training. On the other hand, building the equality constraint between spikes and continuous camera poses better utilizes the continuity of spikes. By accumulating spikes and rendering images in SpikeGS in series, the exposure-like imaging equation is achieved for training. Secondly, we propose a very simple but effective mapping network *Spike-based Instant Mapping (SIM)* from spikes to instant images to support the *Instantaneous Imaging*, to offer reliable supervision signals for rendering SpikeGS. SIM is simply composed of several convolutional layers and incorporates blind spots to enable self-supervised training through spike firing frequency. Our SIM achieves an ultra-lightweight (30K params) design with a very fast inference speed (>1200FPS). In addition, we generate a high-quality spike-based 3D dataset for training and validation. Experiments demonstrate the superior 3D scene reconstruction capabilities of SpikeGS, proving the potential of spiking cameras in 3D vision. The contributions of this work can be summarized as follows:

- We make the first attempt to introduce the 3D Gaussian Splatting (3DGS) with spike cameras in high-speed capture, and constructing **SpikeGS**.
- To train SpikeGS efficiently and effectively, we establish computational equations that relate the rendering process of 3DGS to the instantaneous imaging and exposure-like imaging processes of continuous spikes.
- We establish a very lightweight but effective mapping process from spikes to instant images to assist training.
- Experiments demonstrate the superior 3D scene reconstruction capabilities of SpikeGS on existing and our proposed datasets.

## 2 RELATED WORKS

### 2.1 Spike-based Image Reconstruction

Spike cameras [9], as a novel type of bio-inspired camera, feature the capability of emitting spike bit stream with extremely low latency, thus endowing spike cameras with substantial advantages in the realm of the high-speed image reconstruction area. Specifically, Zhu et al. [45] initially proposes a straightforward spike-based reconstruction method "texture from play-back (TFP)", which closely aligns with the imaging principles of conventional cameras. Inspired by the spike camera's biological principles, studies like [43, 44] have employed short-term synaptic plasticity and retinal imaging principles to transform the spike stream into a high frame rate video sequence. However, these approaches often suffer from significant image quality degradation in real-world scenarios due to inadequate modeling of spike noise. Addressing this, Zhao et al. [40] and Zhang et al. [39] leveraged the powerful nonlinear fitting capabilities of CNNs to train an end-to-end model for converting the spike stream into sharp images on synthetic datasets. While supervised methods trained on the synthetic dataset suffer from significant performance degradation when applied to real-world scenarios, Chen et al. [4] constructed a self-supervised spike-based reconstruction framework that jointly predicts optical flow and grayscale images. Some works focus on color spike cameras [6] or deblurring reconstruction [5].



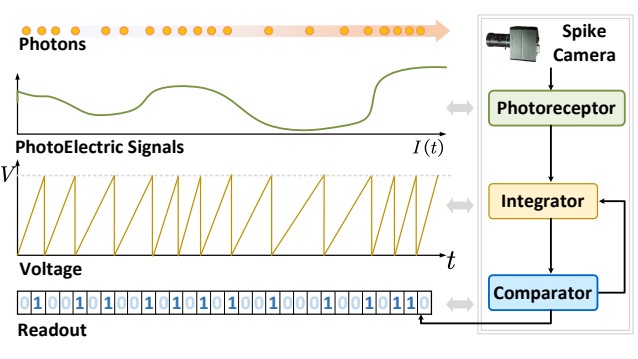

Figure 2: Illustration of the principle of spike cameras.

## 2.2 Novel View Synthesis

**Neural Radiance Fields** Since the introduction of Neural Radiance Fields (NeRF) [24], which represents scenes implicitly and constructs one differentiable 3D scenes reconstruction framework, a significant amount of research has been garnered [15, 20, 34]. However, the quality of 3D scenes reconstructed by NeRF significantly deteriorates when the input image quality is severely degraded, such as in cases of motion blur. To this end, recent studies resort to bio-inspired event cameras, which output events with low temporal latency. For instance, Klenk et al. [11] utilized an event camera and established the E-NeRF framework, which can recover sharp scenes from events under high-speed camera movement. Rudnev et al. [29] learned the 3D RGB representation using a color event camera. Low and Lee [19] established a real event generation physical model and proposed Robust e-NeRF, capable of reconstructing high-quality scenes from sparse and noisy events produced by non-uniform moving cameras. Qi et al. [28] leveraged the complementary information between event and blurry images. Some studies[1, 21] focus on constructing dynamic NeRFs, *i.e.*, utilizing events to recover dynamic scenes with rigid transformations, which is challenging for traditional cameras owing to the limited frame rates.

**3D Gaussian Splatting** Kerbl et al. [10] proposes the novel real-time radiance field rendering approach with the 3D Gaussian splatting which has recently become a potent tool in computer graphics and vision. Scenes are represented with 3D Gaussians whose anisotropic covariance is optimized by gradients. Fu et al. [8] utilized geometric information and continuity in the video to get rid of the Structure-from-Motion(SfM) preprocessing. Yu et al. [38] emphasizes the importance of frequency constraints in 3DGS to avoid artifacts when sampling rates vary. Some works have been proposed to deal with the blurry images led by camera motion [14, 26, 30, 42]. Deblurring 3DGS [14] consists of a small network predicting the covariance offset of 3D Gaussians which represents the blur level of the image. BAD-Gaussian [42] models the physical process of motion blur by optimizing the camera trajectory with the exposure time. BAGS [26] models blur by a Blur Proposal Network (BPN) capable of predicting kernels and masks that indicate the blur region and types. Seiskari et al. [30] proposed to utilize the physical image formation process and velocities to incorporate rolling-shutter and motion blur effects.

However, limitations persist regarding the speed of camera motion across different scenes. The exposure photography principle

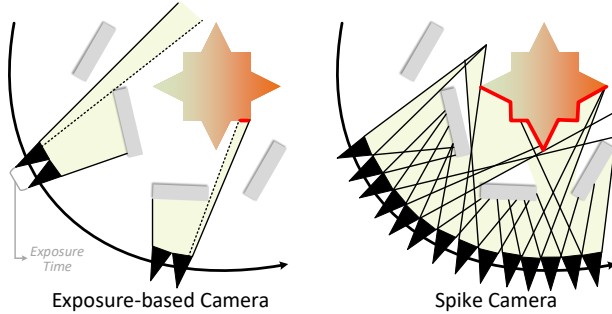

Exposure-based Camera          Spike Camera

Figure 3: An example of why spike cameras possess potential on 3D scene understanding. The green area indicates the visible area. *Left*: The frame-based camera captures discretely with the exposure window, which may lead to blind areas of the scene. *Right*: The spike camera continuously records the scene, which offers more clues for the complex scene.

inherent in traditional cameras also hampers 3DGS-based model performance. For the first time, we introduce the use of spike cameras, leveraging their advantage of ultra-high-speed continuous imaging to effectively model 3D scenes.

## 3 METHOD

### 3.1 Preliminary

**Principle of the Spike Camera**. In the spike camera, each pixel is equipped with a photoreceptor that receives photons at a high frequency, as shown in fig. 2. The arrival of photons alters the photoelectric signals of the receptor sensor and there is an integrator continuously accumulating the voltage. This accumulation continues until the voltage $V$ reaches a predefined threshold $\Theta$. At this moment $t_e$, the pixel emits a spike, and the voltage of the integrator is reset to zero, mathematically formulated as follows:

$$V(t) = \int_{t_s}^{t} \sigma \cdot I(t)dt \,\text{mod}\Theta, \qquad (1)$$

where $I(t)$ represents the instant light intensity at time $t$, $t_s$ is the moment when the previous spike was emitted, and $\sigma$ is the constant photoelectric conversion coefficient. The emitted spike $S$ will be read out at extremely short and uniform intervals $\tau$ ($25\mu$s), which can be formulated as:

$$S_{x,y,k} = \begin{cases} 1, & \text{if } \exists t \in ((k-1)\tau, k\tau], \ V_{x,y}(t) = 0, \\ 0, & \text{if } \forall t \in ((k-1)\tau, k\tau], \ V_{x,y}(t) > 0, \end{cases} \qquad (2)$$

where $(x, y)$ is the pixel coordinate on the imaging plane and $k$ is the $k$-th readout of spikes.

**3D Gaussian Splatting**. 3D Gaussian Splatting stands out as a sophisticated point-based method for 3D scene reconstruction, offering notable advancements beyond the capabilities of Neural Radiance Fields. The core of 3DGS lies in its utilization of a series of 3D Gaussian primitives $\{\mathcal{G}_n | n = 1, ..., N\}$ to encapsulate the scene's spatial attributes.

Each Gaussian primitive is anchored by the central point $\mathbf{p}_n$ and shaped by the covariance matrix $\Sigma_n$, which shape the Gaussian's

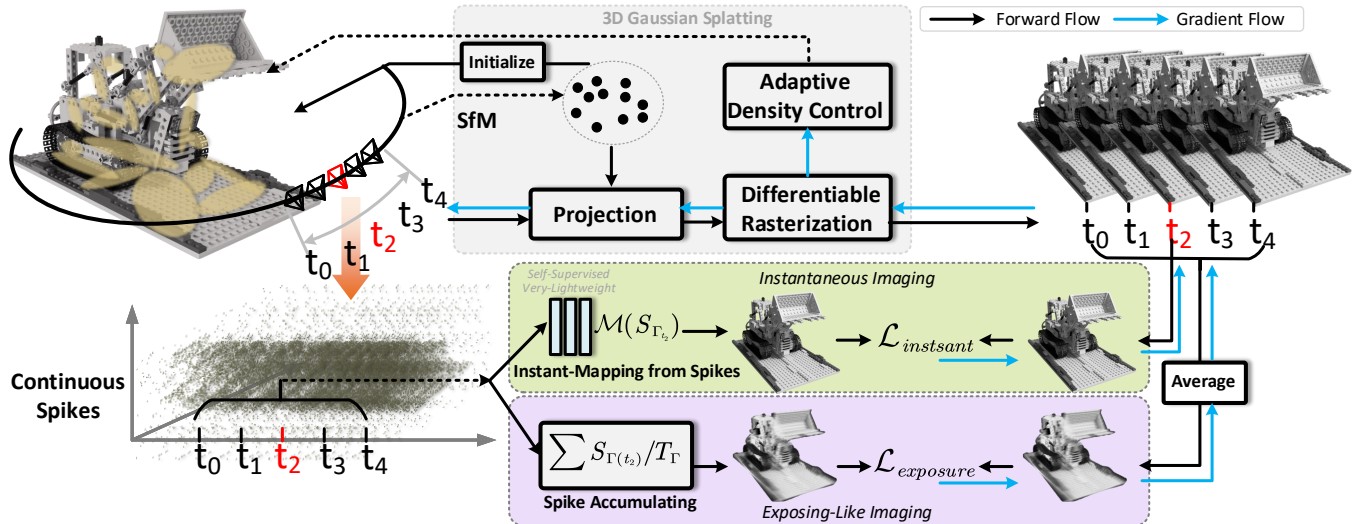

Figure 4: The schematic diagram of our SpikeGS. Combining the *Instantaneous Imaging* and *Exposing-Like Imaging* in spikes, together with the self-supervised lightweight mapping module from spikes to instant images, SpikeGS can be trained effectively.

influence at any selected point **v** in 3D space, described mathematically as:

$$\mathcal{G}_n(\mathbf{v}) = e^{-\frac{1}{2}(\mathbf{v}-\mathbf{p}_n)^T \Sigma_n^{-1}(\mathbf{v}-\mathbf{p}_n)}. \qquad (3)$$

In the rendering phase, these 3D Gaussians are projected onto a 2D plane along the ray $r$, resulting in 2D Gaussian forms $\mathcal{G}_n^{2D}$. Throughout this process, the 3D Gaussians are endowed with additional properties, such as opacity $\alpha$ and color $c$, which play a crucial role in the rendering equation:

$$C(r) = \sum_{k=1}^{N} T_i \alpha_i \mathbf{c}_i \mathcal{G}_i^{2D}, i = D_k, T_i = \prod_{j=1}^{i-1}\left(1 - \alpha_j \mathcal{G}_j^{2D}\right), \qquad (4)$$

where $D$ represents the index of the 3D Gaussian primitives set rearranged according to their depth over the rendered tile.

## 3.2 Analysis on Spike-based Views

***Potentials on Continuous Imaging in 3D Scenes***. In high-speed motion settings, using traditional cameras for 3D reconstruction faces two challenges: 1) insufficient frame rates of traditional cameras lead to missed details due to occlusion in the scene and the camera at certain viewpoints; 2) images captured by traditional cameras in high-speed scenarios tend to blur, as in Fig. 3(*left*). The spike camera offers a solution by outputting the continuous spikes at 40,000 Hz with minimal latency, which ensures that the full details of the captured object are visible even under high-speed camera motion settings, as in Fig. 3(*right*). Recent spike-based image reconstruction methods [39, 40] have demonstrated the capability to recover sharp images from spikes at any given timestamp.

## 3.3 Spike-based Gaussian Splatting

We aim to train a high-quality 3DGS with spikes as supervision. During recording the scene with the spike camera, spikes are continuous. Thus, at training, a 3DGS model can be denoted as:

$$\hat{I}(t) = \mathcal{F}_{3dgs}(\mathcal{P}C, v(t)) \qquad (5)$$

where $\mathcal{P}C$ is the initial points, $v(t)$ is the camera pose at some timestamp $t$ and $\hat{I}(t)$ is the rendered image. Spikes are irregularly binary data, which can be viewed directly. Then the key problem in spike-based 3DGS is raised:

***how to deal with the supervision for $\hat{I}(t)$ at view $v(t)$?***

To supervise the training of continuous 3D scenes based on the discrete spike stream, a straightforward idea is to construct a virtual exposure window as in traditional RGB cameras, *i.e.*, accumulating a large number of spikes and summing up them to get the image. However, in the experimental setting of this paper, the camera moves around the scene at extremely high speeds, leading to significant motion blur in the images obtained through this virtual exposure method. In our SpikeGS, we focus on two factors to assist the training of high-quality 3D scenes from the fast-moving spike camera: (A) *Instantaneous Imaging from spikes*, (B) *Exposing-like Imaging from spikes*. The SpikeGS framework is shown in Fig. 4.

**(A) Instantaneous Imaging from spikes.** In 3DGS, it generally follows the assumption of instantaneous exposure where the images for supervision should denote the instant light intensity. To address this, an **ideal** mapping $\mathcal{M}$ from spikes to instant images is essential, as follows:

$$\bar{I}(t) = \mathcal{M}\left(S_{\Gamma(t)}\right), \qquad (6)$$

where $S_{\Gamma(t)}$ is a segment of spikes in a time interval $\Gamma$ around the $t$, and $\bar{I}(t)$ is the instant image at $t$ predicted from spikes.

Benefit from the continuity of spikes, plenty of motion and texture information are contained in the spikes. If the mapping $\mathcal{M}$ can be established, the instant imaging loss can be formulated as follows to offer SpikeGS supervision as in Fig. 4 for training:

$$\mathcal{L}_{instant} = \|\hat{I}(t) - \bar{I}(t)\|_1. \qquad (7)$$

**(B) Exposing-like Imaging from spikes.** In our setting, a spike camera records a 3D scene continuously, indicating that the camera poses are also non-uniformly continuous. In the time interval $\Gamma(t)$

 

around the view $v(t)$ at $t$ (the duration of $\Gamma$ is denoted as $T_\Gamma$), spike streams and camera poses are both continuous. We aim to build the mathematical formulation between spikes and camera poses. For camera poses, 3DGS itself inputs camera poses and outputs the rendered image at the corresponding view. Assuming the output of 3DGS is an instant clear image, then the mean of its rendering results with continuous poses will approximate an exposure-like blurred image $\hat{B}_t$ with an exposure time of $T_\Gamma$, as follows:

$$\hat{B}(t) = \frac{1}{T_\Gamma} \int_{t-T_\Gamma/2}^{t+T_\Gamma/2} \mathcal{F}_{3dgs}(\mathcal{PC}, v(t)). \tag{8}$$

However, in real-world data capturing, the camera pose cannot be read out at any timestamps. They are recorded discretely. Suppose that there are $K$ poses in $T_\Gamma$, then Eq.8 can be re-write as:

$$\hat{B}(t) = \frac{1}{K} \sum_{k=0}^{K} \mathcal{F}_{3dgs}(\mathcal{PC}, v(t_k)). \tag{9}$$

In the spike stream, the exposure-like image in the $\Gamma(t)$ can be achieved by accumulating spikes along the time axis. with the characteristics of spikes, we can formulate an approximate equation between poses and spikes aiming to train the SpikeGS:

$$\mathcal{L}_{exposure} = \|\hat{B}(t) - \frac{S_{\Gamma(t)}}{T_\Gamma}\|_1. \tag{10}$$

The illustration is shown in Fig. 4. In this way, utilizing the equation Eq. 7 and Eq. 10, the SpikeGS can be trained.

## 3.4 Ideal Mapping of Spikes to Instant Image

For the **instantaneous imaging from spikes**, recall that to successfully achieve the training of SpikeGS, an ideal mapping $\mathcal{M}$ from the segment of spikes to instantaneous image (as in Eq. 6) is needed to satisfy the supervision loss in Eq. 7. Thus, in this section, we are dedicated to dealing with the problem:

### How to get the ideal mapping $\mathcal{M}$ from spikes?

*What is an ideal mapping from spikes to images?* (1) the mapping should be **High-Quality** and **Generalized** in the 3D scene, which means that the image recovered from spikes has sharp textures across all the views. (2) To meet the feature of real-time rendering and very-fast training of the 3DGS, the mapping should be **Simple but Effective**. Upon these requirements, we build a new **Spike Instant Mapping (SIM)** network ($\mathcal{M}(\cdot)$).

Several approaches have been proposed to recover sharp images from spike segments. Although TFI and TFP [45], as the most basic and fast numerical analysis spike reconstruction algorithms, can recover images fast, their quality is poor with noise and blurring. Some supervised deep learning-based methods proposed to recover high-quality grayscale images from spike streams rely on training on large synthetic datasets. The capability of generalization is poor and the model is also complex and has a low speed of inference.

In SpikeGS, from the perspective of generalization ability, we aim to accomplish the mapping $\mathcal{M}(S_{\Gamma_T})$ in a self-supervised manner, training the specific model with spikes in each 3D scene.

SSML [3] is the first self-supervised reconstruction algorithm tailored for spike cameras. It adopts a blind spot network (BSN) structure [13, 16, 17, 35, 36] to predict the current pixel from neighboring spikes. However, SSML's network structure and computing

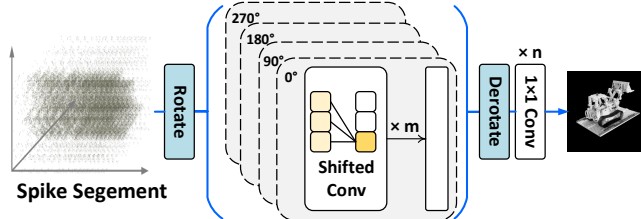

**Figure 5: Model Architecture for mapping spikes to images.**

processes are relatively complex, and its training cost is lagging compared to the fast 3DGS process. Therefore, we aim to build a 'simple but effective' self-supervised module and strive to achieve a lightweight design, motivated by BSN.

**Principle of BSN.** The BSN was initially utilized in self-supervised image-denoising tasks. It relies on the assumption of Noise2Void [12], which assumes that under the premise of noise mean being zero and noise having no spatial correlation, the optimization objective of self-supervised denoising is approximately equivalent to supervised denoising, namely:

$$arg\,min\,\mathbb{E}\{(f_\theta(x) - x)^2\} \approx arg\,min\,\mathbb{E}\{(f_\theta(x) - y)^2\}, \tag{11}$$

where $f_\theta(\cdot)$ represents the denoising network, $x$ denotes the noisy input and $y$ represents the potential sharp image. The equation between the $x$ and $y$ is:

$$x = y + m, \tag{12}$$

where $m$ is the noise. To avoid the identity mapping of the noisy image itself. BSN is thus introduced to address this issue, where the receptive field of each pixel does not include the pixel itself, preventing the identity mapping of the noise.

The spike stream mainly suffers from stochastic thermal noise. Accumulation of dark current can lead the accumulator to reach the firing threshold prematurely, resulting in unexpected binary spike noise. Methods like TFP [45] can construct a low-quality image from spikes with noise, as $I_{noisy}(t) = \mathbf{TFP}(S_{\Gamma(t)})$. Then, between the desired clean image $\bar{I}(t)$ and the $I_{noisy}(t)$, the formulation like Eq: 12 can be established as:

$$I_{noisy}(t) = \bar{I}(t) + m, \tag{13}$$

Thus, for self-supervised BSN-based module $\mathcal{M}$, the optimal module $\mathcal{M}^*$ can be obtained by:

$$\mathcal{M}^* = arg\,min\,\mathbb{E}\{(\mathcal{M}(S_{\Gamma(t)}) - I_{noisy}(t))^2\}. \tag{14}$$

With such a definition, we can use the results of TFP [45] from spikes as the self-supervision for spike-to-image mapping $\mathcal{M}$.

**Constructing the Lightweight Mapping Module for Spikes.** Following the assumptions in SSML [3], the current pixel value can be inferred from the neighboring spike stream. By excluding the central pixel position, the network is unable to learn the noise value at the current position from the spike stream. Fig.5 illustrates the spike reconstruction network we designed. Specifically, we employ a blind spot construction scheme similar to SSML, using shift-based convolutions. We propose to design the **Spike Instant Mapping (SIM)** network ($\mathcal{M}(\cdot)$) most simply with only sequential Conv layers, as shown in Fig. 5.

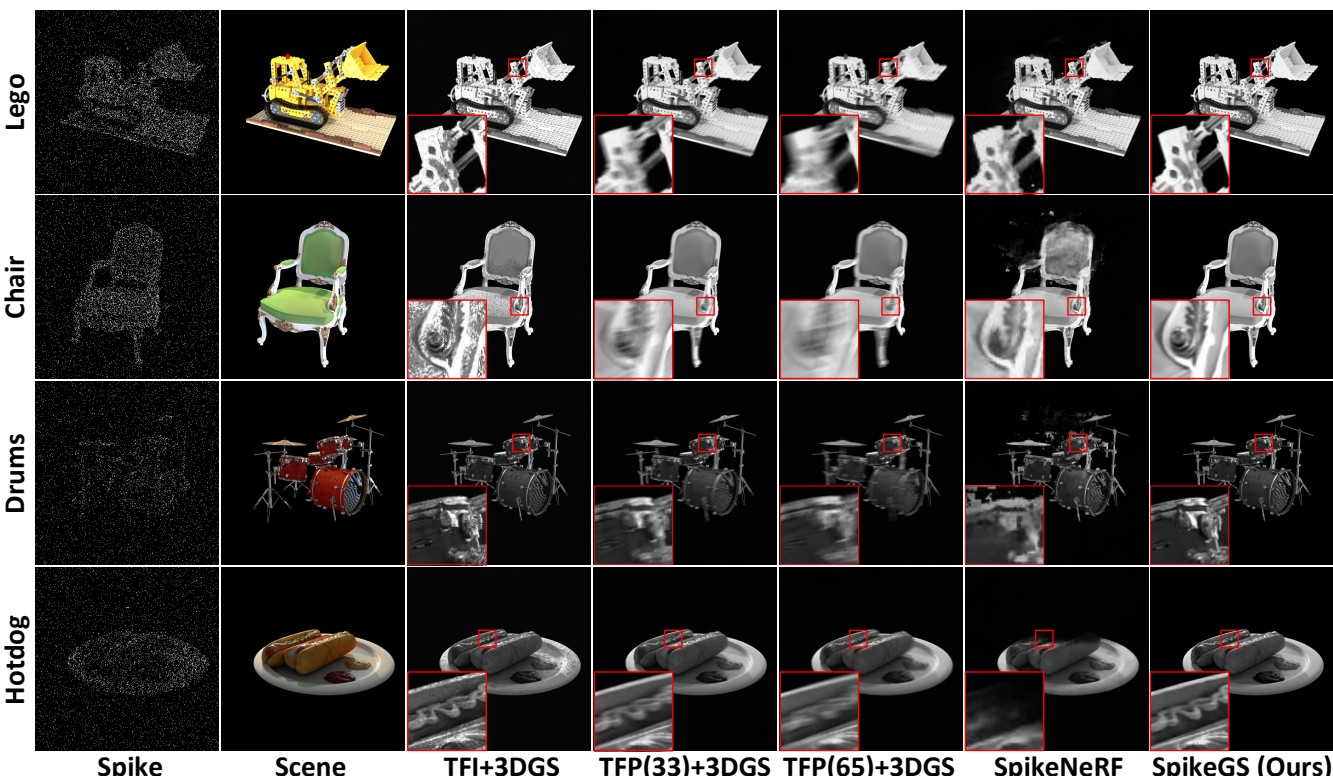

**Figure 6: Qualitative results compared with other methods on the synthetic dataset. We compare with SpikeNeRF [46], and three baseline methods that are the cascading two-stage model TFP(33)+3DGS, TFP(33)+3DGS and TFI+3DGS.**

Since shift-based convolutions cause the network's receptive field to grow in a single direction, we rotate the input spike stream into four parts to obtain a complete receptive field in four directions. The rotated spike stream is fed into $m$ layers of shift-based $3 \times 3$ convolutional layers In this way, the network possesses a receptive field with a $(2m+1, 2m+1)$ size. In the end, the extracted features are combined through $n$ layers of $1 \times 1$ convolutions and the re-rotation operation to obtain the mapped clean image.

Since spikes reach a time resolution of 40,000 Hz, we can safely assume that small pixel displacements are caused by motions within a segment of spikes during a very short interval in $T_\Gamma$. Thus, the $(2m + 1, 2m + 1)$ receptive field is enough when $m$ is small. In our implementation, we only use $m = 3$ and $n = 3$. Thus we construct a very lightweight but robust network for mapping spikes to images. To train the BSN-based $\mathcal{M}$, we simply utilize L1 loss:

$$\mathcal{L}_\mathcal{M} = \|\mathcal{M}(S_\Gamma) - I_{noisy}\|_1 = \|\mathcal{M}(S_\Gamma) - \mathbf{TFP}(S_\Gamma)\|_1. \quad (15)$$

### 3.5 Training the SpikeGS

The loss in Eq. 7 holds with the achievement of idea mapping $\mathcal{M}$ from spikes to images which meets the requirement of Eq. 6. Thus, the loss function for training SpikeGS is as follows:

$$\mathcal{L}_{total} = \mathcal{L}_{instant} + \lambda \cdot \mathcal{L}_{exposure}, \quad (16)$$

where $\lambda$ is a hyperparameter to balance the weight of two losses.

## 4 EXPERIMENTS

### 4.1 Datasets

To our knowledge, there is a related dataset [46] focusing on spike-based 3D reconstruction. However, this dataset has converted the spike stream into image format, rendering it impractical to implement our SpikeGS on this dataset. Therefore, we construct a new synthetic dataset and provide the raw spike stream instead of the image format, as described in Zhu et al. [46].

***Synthetic Dataset.*** To evaluate the quantitative performance of our approach, we first conduct experiments on synthetic scenes provided by Mildenhall et al. [24]. We begin by designing a virtual camera path in Blender that orbits and ascends in a spiral manner around the captured scene. Following this, we render the video sequence along the designed camera path and employ the XVFI frame interpolation algorithm [31] to generate 7 additional frames between each pair of adjacent frames. Finally, we turn to a physically based spike simulator [41], which adopts the Poisson model into the spike simulation process, to convert the high-frame-rate video sequences into the spike stream with low latency.

***Real-world Dataset.*** We evaluate the performance of SpikeGS on the real-world spike dataset released by Zhu et al. [46]. This dataset is captured utilizing a spike camera with a spatial resolution of 250×400. Five real-world scenarios are recorded, each comprising 35 spike images captured by the fast-moving spike camera. The

**Table 1: Quantitative Results on the built Synthetic dataset.**

| Method | Lego | Chair | Materials | Drums | Mic | Hotdog | Ficus |
|---|---|---|---|---|---|---|---|
| | PSNR/SSIM ↑ | PSNR/SSIM ↑ | PSNR/SSIM ↑ | PSNR/SSIM ↑ | PSNR/SSIM ↑ | PSNR/SSIM ↑ | PSNR/SSIM ↑ |
| TFP(33)+3DGS | 26.03/89.18 | 28.65/95.45 | 29.17/93.85 | 25.76/91.97 | 31.03/95.80 | 33.36/96.43 | 30.12/96.31 |
| TFP(65)+3DGS | 23.19/84.55 | 25.54/93.11 | 26.49/91.24 | 24.12/89.23 | 29.04/94.1 | 30.82/95.11 | 26.71/93.60 |
| TFI+3DGS | 24.52/86.33 | 25.12/87.99 | 25.12/87.99 | 27.04/93.08 | 30.88/95.36 | 25.97/88.05 | 29.54/94.81 |
| SpikeNeRF | 19.24/89.77 | 19.63/90.23 | 25.48/94.34 | 22.07/89.91 | 29.47/95.39 | 23.18/93.62 | 25.42/95.30 |
| SpikeGS (**Ours**) | **32.67/96.44** | **35.53/98.53** | **34.18/97.40** | **28.82/96.07** | **34.80/98.29** | **37.39/98.05** | **36.81/99.00** |

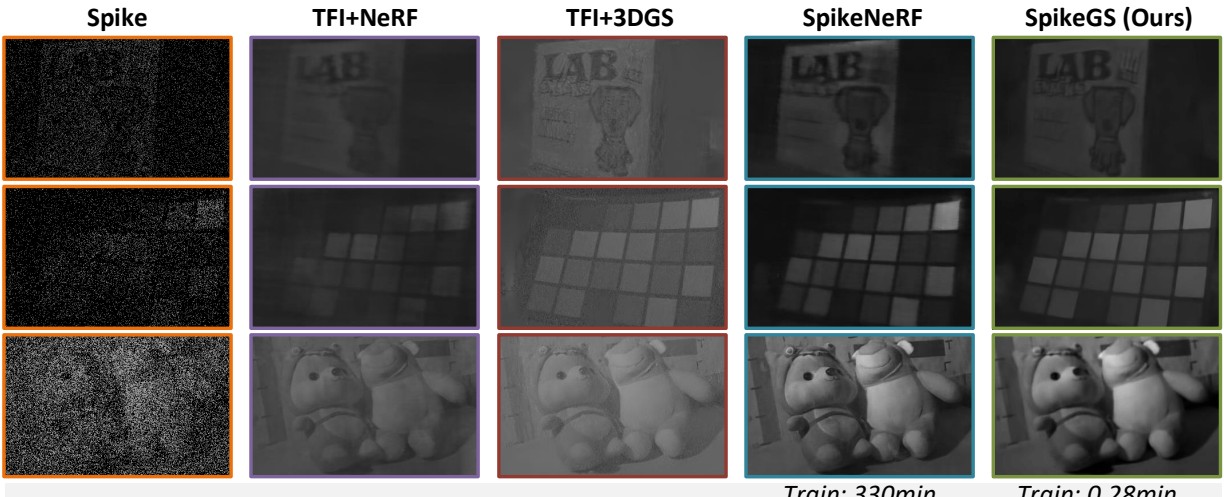

Figure 7: Real-World Comparison with other methods on the dataset in SpikeNeRF [46]. We mainly compare with SpikeNeRF, and the two baseline methods are the cascading two-stage model TFI+NeRF and TFI+3DGS.

dataset is organized in the LLFF [23] format, slightly different from the 360 synthetic scenes constructed in the synthetic dataset.

## 4.2 Training Details

All experiments were conducted on a single NVIDIA GTX 4090, with PyTorch. SpikeGS is trained for 30k iterations taking about 15 minutes, and the learning rate and scheduler settings are identical to those of standard 3DGS. In the implementation, the number of continuous camera poses $K = 5$ used for calculating $\mathcal{L}_{exposure}$, corresponds to a spike stream length of 33. During training, the parameters of the self-supervised network $\mathcal{M}$ were frozen to provide the supervisory signal for $\mathcal{L}_{instant}$.

## 4.3 Quantitative and Qualitative Comparison

We compare our SpikeGS with the SpikeNeRF [46], the only spike-based 3D reconstruction work to our knowledge, in the synthetic and real-world scenarios for quantitative and qualitative comparisons. As for the baseline methods, we chose two direct spike-to-image reconstruction methods, TFI [45] and TFP [45](window size = 33 and 65), and cascaded them with standard 3DGS [10], completing the comparison using the two-stage approach. We choose them to thoroughly show the effectiveness of the 3DGS and our proposed self-supervised BSN network. In the following, We compare our

SpikeGS against other methods mainly from two aspects: image quality and training speed.

*Synthetic Results*. In Tab. 1, we present the results of our method compared to others across all 7 scenes of the synthetic dataset. PSNR and SSIM are used as the quantitative metrics. The results show that the quality of novel view synthesis by SpikeGS significantly surpasses other methods. Specifically, compared to those that use TFI and TFP as cascading modules with 3DGS as the baseline model, SpikeGS surpasses them by approximately 5.3dB, 7.7dB, and 7.5dB in PSNR, respectively. Meanwhile, compared to SpikeNeRF, SpikeGS exceeds by more than 10.8dB. This indicates the high effectiveness of the SpikeGS in 3D reconstruction from spikes. Fig. 6 provides a comparison of visual results. As shown in the results, the images predicted by SpikeNeRF exhibit blurring effects as well as poor adaptability to the motion-induced spike stream; in contrast, images predicted by SpikeGS maintain clear textures and smooth edges, with enhanced realism. Moreover, the training time of SpikeNeRF is about 10 hours (**600min**), while SpikeGS only needs **15min** for training.

*Real-world Results*. Regarding training speed, our SpikeGS demonstrates a substantial efficiency gain against other methods as evidenced in Fig. 7. SpikeGS completes the training in merely 0.28 minutes, a significant reduction from the 330 minutes required by

**Table 2: Ablation Study on Losses in Modules in the SpikeGS.**

| Method | Lego | Chair | Materials | Drums | Mic | Hotdog | Ficus |
|---|---|---|---|---|---|---|---|
| | PSNR/SSIM ↑ | PSNR/SSIM ↑ | PSNR/SSIM ↑ | PSNR/SSIM ↑ | PSNR/SSIM ↑ | PSNR/SSIM ↑ | PSNR/SSIM ↑ |
| Only $\mathcal{L}_{exposure}$ | 29.23/92.94 | 31.73/96.91 | 31.42/95.56 | 26.90/94.19 | 32.72/97.34 | 35.48/96.92 | 34.74/98.41 |
| Only $\mathcal{L}_{instant}$ | 32.28/96.30 | 34.62/98.42 | 33.62/97.17 | 28.09/96.01 | 34.22/98.09 | 36.11/97.87 | 36.47/98.95 |
| $\mathcal{L}_{exposure}$ + $\mathcal{L}_{instant}$ | **32.67/96.44** | **35.53/98.53** | **34.18/97.40** | **28.82/96.07** | **34.80/98.29** | **37.39/98.05** | **36.81/99.00** |

SpikeNeRF. This marked decrease in training time, by over three orders of magnitude, is mainly attributed to the employment of the 3DGS and our designed extremely lightweight BSN, which has faster speed compared to the NeRF framework and SNN as in SpikeNeRF. In terms of image quality, the presented visual outputs indicate that SpikeGS maintains higher reconstruction fidelity than other methods. Specifically, the 'Box' and 'Grid' examples, which are typically challenging due to their regular geometries and uniform patterns, are rendered with greater clarity and less noise by SpikeGS. Moreover, the 'Dolls' instance, characterized by intricate textures and shading, appears to be reconstructed with greater fidelity, indicating the superior handling of subtle image features by our SpikeGS.

## 5 ABLATION STUDY

### 5.1 Ablation on Modules

We conduct ablation experiments on the two types of loss, specifically $\mathcal{L}_{insant}$ and $\mathcal{L}_{exposure}$. The results in Tab. 2 show that if each loss is adopted individually, the model performance decreases. When they are trained together, the model performance significantly increases by 2.57dB and 0.68dB, respectively. This experiment demonstrates the rationality and effectiveness of our architectural design. Through exploring two types of imaging characteristics of spikes, SpikeGS is efficiently and effectively trained.

### 5.2 Ablation on Continuous Rendering Loss

**Table 3: Abation study on the Continuous Rendering Loss with different lengths of spikes for supervision.**

| Render Images | Spikes | PSNR ↑ | SSIM ↑ | Train |
|---|---|---|---|---|
| 13 | 97 | 33.97 | 97.39 | 25.5min |
| 9 | 65 | 34.28 | 97.66 | 22.1min |
| 5 | 33 | 34.31 | 97.68 | 15.0min |

We utilize the average firing rate of the continuous spike stream as the supervision for the Continuous Rendering Loss, simulating the real long-exposure process to optimize the pixel distribution of images with only the short-exposure imaging loss. We conduct ablation experiments on the number of continuous rendering images and the length of the spikes for the loss. The results from Tab. 3 indicate that excessively long exposure times lead to a certain degree of performance degradation and increased training time. Therefore, We ultimately adopt the setting of rendering 5 images.

**Table 4: Performance of the lightweight self-supervised reconstruction model.**

| Scene | PSNR ↑ | SSIM ↑ | Speed | Param |
|---|---|---|---|---|
| Lego | 33.36 | 96.12 | | |
| Chair | 35.45 | 98.13 | | |
| Materials | 37.24 | 98.70 | | |
| Drums | 32.10 | 97.00 | 1200+FPS | **30K** |
| Mic | 35.07 | 98.01 | | |
| Hotdog | 37.64 | 97.30 | | |
| Ficus | 39.28 | 99.08 | | |

### 5.3 Discussion on Reconstruction Model

In this section, we aim to highlight and analyze the advantages and significance of our hyper-quantized self-supervised reconstruction model, tinySpkRecon. (1) In the context of 3D scene understanding based on spike cameras, there is no image information available for designing reconstruction models with supervised learning, and supervised models often lack generalizability; Thus, exploring high-performance self-supervised models is essential. (2) Supervised reconstruction models, such as Spk2ImgNet [40], suffer from slow inference time, poor generalizability, and extremely high computational complexity. Complex reconstruction models contradict the real-time rendering characteristics of 3DGS; therefore, designing ultra-lightweight, fast-inference, and highly generalizable self-supervised models is necessary. Tab. 4 demonstrates that our designed self-supervised model performs well in scene reconstruction with strong generalizability; at the same time, our model can infer at **speeds exceeding 1200 FPS** (frames per second) on a single 4090 GPU, with only **30K parameters** required. Such design and performance will not increase any computational burden on the 3DGS pipeline and greatly enhance the rendering performance of our SpikeGS.

## 6 CONCLUSION

We make the first attempt to introduce the 3D Gaussian Splatting (3DGS) with spike cameras in high-speed capture, and constructing **SpikeGS**. A lightweight self-supervised model *tinySpkRecon* is proposed for recovering images from spikes. The loss combined with *Instantaneous imaging* and *Exposure-like imaging* is designed to improve rendering quality. Experiments demonstrate the superior 3D scene reconstruction capabilities of SpikeGS both on the synthetic and the real-world datasets.

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
