# OpenReview forum: "SpikeGS: 3D Gaussian Splatting from Spike Streams with High-Speed Camera Motion"
_acmmm.org/ACMMM/2024/Conference — MM2024 Poster_

### Official Review · Reviewer_L4gM · 2024-05-08

**Rating:** 4
**Confidence:** 3

**Summary:**

The paper employs the 3DGS algorithm for the first time in the data stream of spike camera to address the time-consuming issue of existing rendering reconstruction algorithms, which demonstrates sufficient innovation and reasonable motivation. Simultaneously, equations are established to compute the rendering process of 3DGS and the instantaneous imaging and exposure-like imaging processes of continuous spike streams. A highly lightweight network is utilized to support rapid mapping. The effectiveness of the algorithm is verified through synthetic data and real data separately, with comprehensive and adequate experimental setups.

**Strengths:**

1. The first application of the 3D Gaussian Splatting algorithm demonstrates strong innovation and validates its suitability for utilization in spike streams.

2. Detailed derivations of equations for the rendering process of 3DGS and the instantaneous imaging and exposure-like imaging are provided, serving as a reference basis for future work.

**Limitations:**

1. The performance of the algorithm on real data is not sufficiently good. Why not utilize reference-free evaluation methods to provide quantitative comparisons? In the "Toys" instance in Figure 7, there appears to be significant blur in your results, a point overlooked in the analysis.

2. The writing details of the paper need to be more rigorous. For instance, the unit of the SSIM metric in all tables need to be specified, and the scene names in Figure 7 need to be included.

**Suitability:**

2

---

### Official Review · Reviewer_eUTo · 2024-05-20

**Rating:** 4
**Confidence:** 2

**Summary:**

Novel View Synthesis plays a crucial role by generating new 2D renderings from multi-view images of 3D scenes. However, capturing
high-speed scenes with conventional cameras often leads to motion
blur, hindering the effectiveness of 3D reconstruction. To address
this challenge, high-frame-rate dense 3D reconstruction emerges as
a vital technique, enabling detailed and accurate modeling of realworld objects or scenes in various fields, including Virtual Reality or
embodied AI. Spike cameras, a novel type of neuromorphic sensor,
continuously record scenes with an ultra-high temporal resolution,
showing potential for accurate 3D reconstruction. Despite their
promise, existing approaches, such as applying Neural Radiance
Fields (NeRF) to spike cameras, encounter challenges due to the
time-consuming rendering process. To address this issue, we make
the first attempt to introduce the 3D Gaussian Splatting (3DGS) into
spike cameras in high-speed capture, providing 3DGS as dense and
continuous clues of views, then constructing SpikeGS. Specifically,
to train SpikeGS, we establish computational equations between
the rendering process of 3DGS and the processes of instantaneous
imaging and exposing-like imaging of the continuous spike stream.
Besides, we build a very lightweight but effective mapping process
from spikes to instant images to support training. Furthermore, we
introduced a new spike-based 3D rendering dataset for validation.
Extensive experiments have demonstrated our method possesses
the high quality of novel view rendering, proving the tremendous
potential of spike cameras in modeling 3D scenes.

**Strengths:**

We make the first attempt to introduce the 3D Gaussian
Splatting (3DGS) with spike cameras in high-speed capture,
and constructing SpikeGS.
• To train SpikeGS efficiently and effectively, we establish
computational equations that relate the rendering process
of 3DGS to the instantaneous imaging and exposure-like
imaging processes of continuous spikes.
• We establish a very lightweight but effective mapping process from spikes to instant images to assist training.
• Experiments demonstrate the superior 3D scene reconstruction capabilities of SpikeGS on existing and our proposed
datasets.

**Limitations:**

It's just a simple overlap, and it doesn't feel consistent with the meaning of the theme.

**Suitability:**

2

---

### Official Review · Reviewer_vhCt · 2024-05-23

**Rating:** 3
**Confidence:** 2

**Summary:**

The paper makes significant contributions to the field of 3D reconstruction by introducing a novel method that integrates spike cameras with 3D Gaussian Splatting. This approach addresses the challenges posed by motion blur in high-speed captures and demonstrates the potential for high-quality 3D scene reconstruction.

**Strengths:**

1.	This is the first attempt to introduce 3D Gaussian Splatting (3DGS) into spike cameras, showcasing an innovative approach to high-speed 3D reconstruction.
2.	A new spike-based 3D rendering dataset was introduced for validation, providing a robust benchmark for evaluating the proposed method.
3.	The problem analysis and the solution in each section are clear and easy to follow, making the paper accessible and informative for readers.
4.	The accompanying illustrations are very clear, enhancing the understanding of the concepts and methodologies presented.

**Limitations:**

1.	It would be easier to follow the projection process (Lines 375-376) if some references were added or the mathematical process was directly expressed.
2.	What would the experiment result be if the camera moved around the scene at low speeds? The whole work seems to assume that the camera moves at high speeds. Is this assumption necessary for 3D reconstruction, especially for a stationary object?
3.	Is model M the pre-trained model directly used from other work, or is the whole training process two-stage, with the first stage for the self-supervised model M and the second for training 3DGS?
4.	What is the value of $\lambda$ in the ablation experiments in Table.2? Dose different magnitudes of $\lambda$ have a large impact on the results?

**Suitability:**

3

---

### Official Review · Reviewer_Kfnp · 2024-06-06

**Rating:** 1
**Confidence:** 4

**Summary:**

In response to the slow processing speed and long rendering times associated with using NeRF and traditional camera techniques for spike data, this paper introduces SpikeGS. The aim is to achieve 3D Gaussian projection from Spike data streams through the movement of a high-speed camera. This method leverages spike camera, which continuously records scenes at ultra-high temporal resolution, potentially offering advantages for precise 3D reconstruction. This paper represents the first attempt to apply 3D Gaussian projection techniques to spike cameras to enhance the quality and speed of visual synthesis and 3D reconstruction. The paper's topic and method have a certain level of practical value, but there are still some issues that need to be addressed.

**Strengths:**

-This paper follows the typical structure of a scientific paper, including sections such as introduction, related work, methodology, experiments, discussion, and conclusion. This structure allows readers to gradually understand the motivation, methods, experimental results, and significance of the research.
-The authors not only provide quantitative experimental results but also conduct qualitative comparisons and discussions, including comparative analysis and ablation studies. This in-depth analysis helps readers to fully understand the effectiveness of the method and its potential application scenarios.

**Limitations:**

-In the related work section, the review of literature fails to comprehensively cover the recent key advancements in similar technical domains, especially the latest research in other neuromorphic vision sensors.
-The paper constructs a new synthetic dataset, but lacks discussion on the dataset construction method, diversity, and its impact on the results. It is recommended to increase the discussion on dataset characteristics and further validate the method's effectiveness and generality through comparison with other standard datasets.
-The paper lacks necessary detailed information when describing training details in Section 4.2, particularly in the algorithm and parameter settings of 3D Gaussian Smoothing.
-Although the paper introduces a new approach of applying 3D Gaussian Smoothing to spike cameras, the innovative aspects based on existing technologies need to be more clearly articulated. Especially, more quantitative analysis and comparison are needed to demonstrate the advantages and performance improvements of the proposed method in handling high-speed dynamic scenes compared to existing 3D reconstruction techniques.
-The paper uses a large number of professional terms and abbreviations. It is recommended to add a table to organize them.
-Some figures and diagrams (such as Figure 4 and Figure 6) effectively illustrate the points in the paper, but the resolution and labeling of certain key figures need improvement for better understanding of the specific implementation and effects of the model.
-While applying 3D Gaussian Smoothing technology to spike cameras is a novel attempt, fundamentally, this method still combines existing imaging technologies (3D Gaussian Smoothing and spike camera technology). The paper does not fully demonstrate the fundamental technological breakthroughs or significant theoretical innovations brought about by this combination.

**Suitability:**

1

---

### Meta-Review · Area_Chair_hnRM · 2024-06-30

**Recommendation:** Accept (Poster)
**Confidence:** 3

**Metareview:**

The paper presents a novel approach by introducing 3D Gaussian Splatting (3DGS) from Spike Streams with High-Speed Camera Motion. This novel integration, referred to as SpikeGS, is supported by a new spike-based 3D rendering dataset. Despite a few areas needing further clarification and additional rigorous analysis, the overall contribution and potential impact of the work justify its acceptance.

Strengths:

The paper follows the typical structure of a scientific paper, including sections such as introduction, related work, methodology, experiments, discussion, and conclusion. This structure facilitates a clear and gradual understanding of the research.
The authors provide both quantitative experimental results and qualitative comparisons and discussions. The inclusion of comparative analysis and ablation studies offers an in-depth analysis of the method's effectiveness and potential application scenarios.
This is the first attempt to introduce 3D Gaussian Splatting (3DGS) into spike cameras, showcasing a novel approach to high-speed 3D reconstruction.
A new spike-based 3D rendering dataset is introduced for validation, providing a robust benchmark for evaluating the proposed method.
The paper provides detailed derivations of equations for the rendering process of 3DGS and the instantaneous imaging and exposure-like imaging processes, serving as a reference basis for future work.

Limitations:

The paper seems to assume that the camera moves at high speeds. Clarification on the necessity of this assumption for 3D reconstruction, especially for stationary objects, would be beneficial. Additionally, it would be useful to explore the experiment results if the camera moved at low speeds.
Clarification is needed on whether model M is a pre-trained model directly used from other work or if the training process is two-stage, with the first stage for the self-supervised model M and the second for training 3DGS.
The writing details of the paper need to be more rigorous. For instance, the unit of the SSIM metric in all tables needs to be specified, and the scene names in Figure 7 need to be included.

The paper introduces a highly innovative approach to 3D reconstruction using spike cameras and 3D Gaussian Splatting. The novelty of the method, supported by a new dataset and comprehensive analysis, significantly advances the field. While some areas need further clarification and rigorous analysis, the overall contribution and potential impact of the work justify its acceptance. The authors are encouraged to address the identified limitations in their final revisions to further strengthen the paper.